# ANYGRAPH: GRAPH FOUNDATION MODEL IN THE WILD

## ABSTRACT

The growing ubiquity of relational data structured as graphs has underscored the need for graph learning models with exceptional generalization capabilities. However, current approaches often struggle to effectively extract generalizable insights, frequently requiring extensive fine-tuning and limiting their versatility. Graph foundation models offer a transformative solution, with the potential to learn robust, generalizable representations from graph data. This enables more effective and adaptable applications across a wide spectrum of tasks and domains. In this work, we investigate a unified graph model, AnyGraph, designed to handle key challenges: i) **Structure Heterogenity**. Addressing distribution shift in graph structural information; ii) **Feature Heterogenity**. Handling diverse feature representation spaces across graph datasets; iii) **Fast Adaptation**. Efficiently adapting the model to new graph domains; iv) **Scaling Law Emergence**. Enabling the model to exhibit scaling law behavior, where its performance scales favorably with the amount of data and parameter sizes. To tackle these critical challenges, we build the AnyGraph upon a Graph Mixture-of-Experts (MoE) architecture. This approach empowers the model to effectively manage both the in-domain and cross-domain distribution shift concerning structure-level and feature-level heterogeneity. Furthermore, a lightweight graph expert routing mechanism is proposed to facilitate AnyGraph's fast adaptability to new data and domains. Our extensive experiments on diverse 38 graph datasets have demonstrated the strong zero-shot learning performance of AnyGraph across diverse graph domains with significant distribution shift. Furthermore, we have validated the model's fast adaptation ability and scaling law emergence, showcasing its versatility. We have anonymously released our open-sourced AnyGraph implementation at the following link: https://anonymous.4open.science/r/AnyGraph-FECD.

## 1 INTRODUCTION

The growing ubiquity of relational data in the form of graphs has underscored the pressing need for advanced graph learning models that excel at generalization (Fey et al., 2024; Jin et al., 2020). As real-world applications of graph-structured data continue to proliferate across diverse domains, including social networks, academic networks, transportation systems, and biological networks, the ability of graph learning models to effectively handle distribution shifts and adapt to new graph domains has become increasingly crucial (Zhang et al., 2023; Zhao et al., 2024; Mao et al., 2024). Developing models with robust zero-shot learning performance and fast adaptation capabilities can unlock transformative opportunities for leveraging the rich insights encoded within graph data.

The field of graph learning has seen significant advancements in recent years, largely driven by the power of Graph Neural Networks (GNNs) (Liu et al., 2022; Xiao et al., 2021; Li et al., 2021). However, the state-of-the-art models often fall short when it comes to truly generalizable performance. Existing approaches are heavily reliant on arduous fine-tuning processes, making them ill-equipped to handle the diverse array of graph structures and distributions encountered in real-world applications. This inability to adapt swiftly and seamlessly to novel graph domains poses a critical barrier to the widespread adoption of graph learning technologies. Therefore, addressing this challenge is of high importance if we are to fully harness the transformative potential of graph-based insights.

Inspired by the principles that have driven the development of successful foundation models in understanding vision and language data (Wang et al., 2022; 2023), the concept of a versatile graph foundation model holds immense potential to unlock new frontiers in graph learning. By learning

rich, transferable representations from diverse graph-structured data, such a model can be efficiently adapted to a wide array of graph domains and tasks. However, building an effective and adaptive graph foundation model is not a trivial endeavor. Several key challenges must be overcome, including:

(i) **Structure Heterogeneity**. The development of versatile graph models faces the challenge of accommodating diverse structural properties and data distributions in various graph datasets. For instance, graphs can exhibit substantial heterogeneity in node degree distributions, ranging from homogeneous to highly skewed patterns. Similarly, graph structures can vary greatly in complexity, from simple topologies to intricate, hierarchical arrangements. These structural variations can significantly impact the performance and generalization of graph models. Effectively addressing this heterogeneity is critical for developing unified models that can thrive across diverse graph data.

(ii) **Feature Heterogeneity**. Graphs exhibit substantial heterogeneity in their node and edge features, which can span categorical attributes, continuous numerical data, and multi-modal content. Furthermore, the dimensionality and semantics of these features often vary dramatically across different graph domains. For instance, a social interaction graph may include textual content and demographic information associated with its nodes, while a molecular graph may feature atomic compositions and bond types. Effectively handling this feature heterogeneity is crucial for building a versatile graph model capable of generalizing across diverse graph domains.

(iii) **Fast Adaptation for Broad Applicability**. A key capability for graph foundation models is the ability to efficiently adapt to new graph dataset and domains. Rather than requiring extensive retraining or fine-tuning, the ideal model should be able to quickly adjust its parameters and learning strategies to handle the structural and distributional characteristics of previously unseen graph datasets. By seamlessly generalizing and performing well across a diverse range of real-world scenarios – from user behavior graphs to transportation networks and biological systems – these adaptable models can unlock transformative insights across an ever-expanding universe of graph-structured data.

(iv) **Scaling Laws for Transformative Graph Capabilities**. A key characteristic of successful foundation models in domains like CV (Cherti et al., 2023) and NLP (Muennighoff et al., 2024) is their ability to exhibit scaling laws - where performance systematically improves as the model size or training dataset increases. By harnessing this emergent scaling phenomenon, graph foundation models can unlock unprecedented levels of capability and generalization, far surpassing the limitations of fixed-capacity architectures. As the size of graph datasets and model complexity grow, these scaling-aware designs can continue delivering transformative performance gains.

**The Presented Work**. To tackle the above challenges, our AnyGraph model is built upon a Mixture-of-Experts (MoE) architecture, which allows for effective handling of both the in-domain and cross-domain distribution shift in structure-level and feature-level. The proposed graph MoE paradigm empowers AnyGraph to learn a diverse ensemble of graph experts, each tailored to specific structural characteristics. This enables the model to effectively manage the distribution shift in graph topologies. Furthermore, the MoE architecture facil-

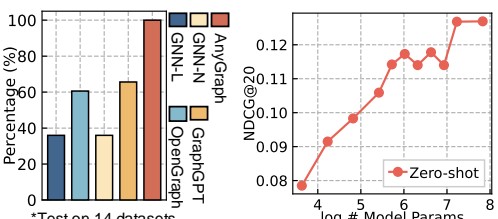

Figure 1: The zero-shot generalizability (left) and scaling law (right) of AnyGraph model.

itates fast adaptation of AnyGraph. Rather than relying on a single, fixed-capacity model, the Graph MoE can efficiently tailor some of its expert networks to capture distinct characteristics of new graph data. A lightweight graph expert routing mechanism also allows AnyGraph to quickly identify and activate the most relevant experts for a given input graph, without requiring extensive retraining or fine-tuning across the entire model. The key findings of this work can be summarized as follows:

- **Methodology Design Motivations of AnyGraph**. Current large graph models (Chen et al., 2024; Liu et al., 2024; Li et al., 2024) often struggle when faced with the substantial heterogeneity found in real-world graph data. This is especially challenging when it comes to feature-level heterogeneity. These fixed-capacity models may encounter interference between different types of graph datasets, and can sometimes overfit to new data, leading to catastrophic forgetting. To address these challenges, the proposed graph MoE architecture was designed with a focus on adaptability. This new paradigm empowers the model to flexibly adjust to the nuances of diverse graph datasets, dynamically selecting the most appropriate experts to learn distinct patterns.

- **Stronger Gernealiation Capacities of AnyGraph**. Through extensive experiments, AnyGraph has demonstrated strong generalization capacities across a wide range of graph tasks and domains. The experimental results showcase the AnyGraph's ability to outperform existing graph models in terms of both predictive performance and robustness to distribution shift.

- **Fast Adapability of AnyGraph**. Our innovative dynamic expert selection mechanism enhances AnyGraph's ability to swiftly adapt to new graph domains. By dynamically routing inputs through relevant experts, AnyGraph can quickly activate the specialized networks best suited for the task. This strong adaptation sets AnyGraph apart from baselines. Evaluation shows its superiority through rapid convergence and exceptional performance, further justifying its cross-domain versatility.

- **The Scaling Law of AnyGraph**. Our experiments reveal that AnyGraph's performance follows the scaling law, where the model continues to improve as model size and training data increase. Additionally, AnyGraph exhibits emergent abilities, where its generalization capabilities see sudden significant improvements with further scaling. This critical scaling law property has been largely overlooked in prior investigations, but it underscores the immense value that AnyGraph derives from its scaling-driven enhancements to generalization performance.

## 2 PRELIMINARIES

**Graph-Structured Data**. A graph $\mathcal{G}$ consists of a set of nodes $\mathcal{V} = \{v_i\}$ and a set of edges $\mathcal{E} = \{(v_i, v_j)\}$. In many cases, each node $v_i$ is associated with a feature vector $\mathbf{f}_i \in \mathbb{R}^{d_0}$. To efficiently utilize such graph-structured data, the link information is typically recorded using an adjacency matrix $\mathbf{A} \in \mathbb{R}^{|\mathcal{V}| \times |\mathcal{V}|}$. Each element $a_{i,j}$ of $\mathbf{A}$ is either 1 or 0, inddicating whether there is an edge from node $v_i$ to $v_j$. Additionally, the feature vectors of the nodes are usually represented by a feature matrix $\mathbf{F} \in \mathbb{R}^{|\mathcal{V}| \times d_0}$, where each row corresponds to a node's feature vector.

**Graph Foundation Models (GFMs)**. The essence of GFMs lies in their strong generalization capabilities. Specifically, a graph foundation model should be able to handle unseen graph data that exhibits significant discrepancies from its training graph datasets. These discrepancies may include differences in feature spaces, as well as variations in node and edge semantics across datasets. Formally, let's denote the training graphs as $\mathbb{S} = \{\mathcal{G}_s\}$, where each graph $\mathcal{G}_s$ is associated with a label set $\mathcal{Y}_s$. Similarly, the set of test graphs is denoted as $\mathbb{T} = \{\mathcal{G}_t\}$, with labels $\mathcal{Y}_t$. With a differentiable training objective $\mathcal{L}$ and an evaluation criterion $\mathcal{C}$ to measure the prediction accuracy of downstream tasks, building a graph foundation model $f_{\mathbf{\Theta}}$ with trainable parameters $\mathbf{\Theta}$ can be formalized as:

$$\arg\max_{f,\mathcal{L}} \sum_{\mathcal{G}_t} \mathcal{C}\left(f_{\mathbf{\Theta}}(\mathcal{G}_t), \mathcal{Y}_t\right), \quad \mathbf{\Theta} = \arg\min_{\mathbf{\Theta}} \sum_{\mathcal{G}_s} \mathcal{L}\left(f_{\mathbf{\Theta}}(\mathcal{G}_s), \mathcal{Y}_s\right) \tag{1}$$

The above formulation reveals that the key to building GFMs are: **i)** the model architecture design ($f$), which must have the capacity to encode diverse feature spaces and structural patterns, and **ii)** the model training process ($\mathcal{L}$), which must effectively traverse such diverse data to find an optimal solution $\mathbf{\Theta}$ for the model $f$. In light of this, our AnyGraph employs a mixture-of-experts architecture with an automated expert routing method, to seamlessly integrate powerful prediction models for highly diverse graph data. AnyGraph is extensively trained on graphs from various applications using multiple featuring methods, with a graph augmentation technique to further enhance data diversity.

## 3 METHODOLOGY

AnyGraph aims to address graph heterogeneity in both structures and node features, while enabling fast adaptation to new data. The proposed graph MoE paradigm enables AnyGraph to learn a diverse ensemble of graph experts, each tailored to specific characteristics. The lightweight expert routing mechanism allows AnyGraph to quickly identify and activate the most relevant experts for a given input graph, without extensive retraining or fine-tuning. Its overall framework is depicted in Fig. 2.

### 3.1 MOE ARCHITECTURE OF ANYGRAPH

**Addressing Cross-domain Graph Heterogeneity**. To model heterogeneous graph patterns across domains, AnyGraph employs a MoE architecture consisting of multiple graph expert models, each responsible for handling graphs with specific characteristics. An automated routing algorithm is

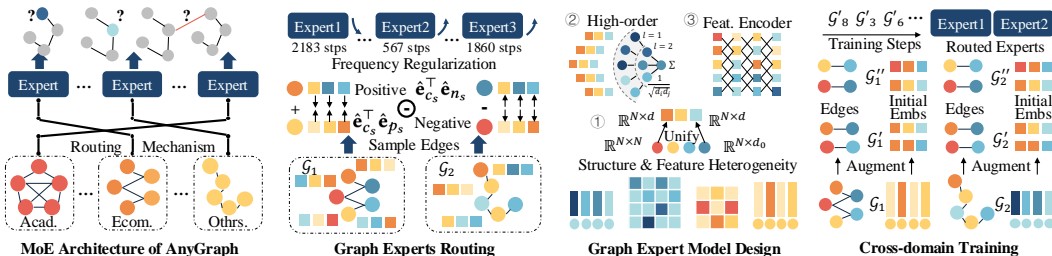

Figure 2: The overall model architecture of the proposed AnyGraph framework.

designed to assign input graph data to the most competent expert model for training and prediction. Specifically, the AnyGraph framework can be denoted as $\mathcal{M} = (f_{\Theta_1}, f_{\Theta_2}, \cdots, f_{\Theta_K}, \psi)$, where $K$ denotes the number of experts. For an input graph $\mathcal{G}$, the routing algorithm $\psi$ firstly identifies the most competent expert model, which is then used for predicting the graph data, as follows:

$$\hat{y}_{i,j} = \hat{\mathbf{e}}_i^\top \hat{\mathbf{e}}_j, \quad \hat{\mathbf{E}} = f_{\Theta_k}(\mathcal{G}), \quad k = \psi(\mathcal{G}) \tag{2}$$

where each expert model $f_{\Theta_k}$ can be viewed as a projection from the graph space to a node embedding space with uniquely trained parameters $\Theta_k$. And $\hat{y}_{i,j}$ represents the dot-product-based prediction of whether the entity $v_i$ should be related to the entity $v_j$. Here, $v_i$ and $v_j$ could be vanilla graph nodes, class labels, or graph labels, enabling link prediction, and node/graph classification tasks.

**Graph Expert Routing Mechanism**. Inspired by the effectiveness of graph self-supervised learning tasks Jin et al. (2022), we propose measuring the competence of expert models on specific graph datasets using the models' self-supervised learning loss values. Specifically, for an input graph $\mathcal{G} = (\mathcal{V}, \mathcal{E})$, the routing mechanism $\psi$ calculates the dot-product-based relatedness scores for some positive edges $(v_{c_1}, v_{p_1}), \cdots, (v_{c_S}, v_{p_S}) \in \mathcal{E}$ and analogously calculates the relatedness scores for some sampled negative node pairs $(v_{c_1}, v_{n_1}), \cdots, (v_{c_S}, v_{n_S}) \notin \mathcal{E}$. The following score difference is then calculated as the competence indicator $\varphi_k$ for the $k$-th expert model regarding the input graph $\mathcal{G}$:

$$\varphi_k = \frac{1}{S} \cdot \sum_{s=1}^{S} \sigma(\hat{\mathbf{e}}_{c_s}^\top \hat{\mathbf{e}}_{p_s} - \hat{\mathbf{e}}_{c_s}^\top \hat{\mathbf{e}}_{n_s}) \tag{3}$$

where $\sigma(\cdot)$ represents the sigmoid activation function, which constrains the competence score to the range of (0, 1). This prevents the few outlier cases where the non-activated score difference is excessively large or small, which could otherwise distort the results.

**Training Frequency Regularization**. Though being empirically accurate in measuring models' competence using the above competence score, this method tends to result in a winner-takes-all sub-optimal situation. In this scenario, a single model, or very few models, is predominantly selected as the most competent expert and is used to handle almost all input graphs. These models generally receive more or better training samples in the early training stages, giving them an advantage over other experts. Consequently, subsequent training samples are also mostly assigned to them due to their performance advantages, ultimately causing other experts to remain largely untrained.

This situation contradicts our motivation of using different expert models to learn different subsets of graph modeling knowledge. To address this, we propose a training frequency regularization approach that recalibrates the competence score as follows:

$$\varphi'_k = \varphi_k \cdot \left( (1 - \frac{m_k}{\sum_{k'} m_{k'}}) \cdot \rho + 1.0 - \frac{\rho}{2} \right) \tag{4}$$

where $\varphi'k$ represents the recalibrated routing score for the $k$-th expert model $f\Theta_k$, based on the number of previously assigned training steps $m_k$ for $k = 1, \cdots, K$. The notation $\rho$ refers to a hyperparameter for the recalibration scale. A larger $\rho$ results in a greater adjustment to the competence score $\varphi_k$. With this additional step, the expert routing mechanism will assign more training instances to the less trained expert models, thereby preventing the aforementioned winner-takes-all situation.

**Fast Adaptation Capabilities of AnyGraph**. With the MoE architecture and routing mechanism, the training and inference process of AnyGraph is conducted by only one expert model. This approach consumes only $1/K$ of the computational and memory resources required for predictions and optimization, compared to other non-MoE graph foundation models based on complex networks like transformers. This enables fast adaptation for AnyGraph when encountering new data.

## 3.2 ADAPTIVE AND EFFICIENT GRAPH EXPERTS

**Addressing In-domain Graph Heterogeneity**. To handle graph data with different adjacency and feature dimensionalities, the expert models of our AnyGraph employ a structure and feature unification process. Adjacency matrices and node features of varying sizes are both mapped into initial node embeddings of fixed dimensionality using a unified mapping function. Inspired by the effectiveness of singular value decomposition (SVD) in extracting important latent features Cai et al. (2023), we utilize SVD for this unified mapping process as follows:

$$\mathbf{U_A}, \Lambda_\mathbf{A}, \mathbf{V_A} = \mathrm{SVD}(\tilde{\mathbf{A}}) \qquad \mathbf{U_F}, \Lambda_\mathbf{F}, \mathbf{V_F} = \mathrm{SVD}(\mathbf{F})$$

$$\mathbf{E}_0 = \mathrm{LayerNorm}\left(\mathbf{U_A}\sqrt{\Lambda_\mathbf{A}} + \mathbf{V_A}\sqrt{\Lambda_\mathbf{A}} + \mathrm{Flip}(\mathbf{U_F}\sqrt{\Lambda_\mathbf{F}})\right) \tag{5}$$

Here, $\mathbf{U_A}, \mathbf{U_A} \in \mathbb{R}^{|\mathcal{V}|\times d}$ and $\mathbf{U_F} \in \mathbb{R}^{|\mathcal{V}|\times d}, \mathbf{V_F} \in \mathbb{R}^{d_0 \times d}$ refer to the $d$-dimensional features obtained through SVD of the Laplacian-normalized adjacency matrix $\tilde{\mathbf{A}}$ and the node feature matrix $\mathbf{F}$, respectively. If the dimensionality of $\tilde{\mathbf{A}}$ or $\mathbf{F}$ is less than $d$, SVD uses a smaller rank $d'$ equal to the smallest dimensionality of $\tilde{\mathbf{A}}/\mathbf{F}$, and the remaining dimensions are padded with zeros up to $d$.

Due to the nature of SVD, the dimensions of these features ($\mathbf{U}_*, \mathbf{V}_*$) are ranked from the most important to the least important, corresponding to the descending eigenvalues in the diagonal matrices $\Lambda_\mathbf{A}$ and $\Lambda_\mathbf{F}$. In light of this characteristic, we propose to better preserve the most important feature dimensions for both $\tilde{\mathbf{A}}$ and $\mathbf{F}$. In particular, the function $\mathrm{Flip}(\cdot)$ reverses the $d$ dimensions of each row for the SVD features of $\mathbf{F}$, such that the important features of $\tilde{\mathbf{A}}$ are aligned with the less important features of $\mathbf{F}$, and vice versa.

**High-order Connectivity Injection**. A non-trainable layer normalization $\mathrm{LayerNorm}(\cdot)$ is applied for numerical stability. The initialized embeddings, denoted as $\mathbf{E}_0 \in \mathbb{R}^{|\mathcal{V}|\times d}$, have consistent representation dimensionality and relatively stable semantics across datasets. To better preserve the multi-hop connection information into the initial embeddings, AnyGraph adopts a simplified GCN without parameters Wu et al. (2019) for $\mathbf{E}_0$ as follows:

$$\mathbf{E}_1 = \sum_{l=1}^{L} \mathbf{E}_0^{(l)}, \;\; \mathbf{E}_0^{(l)} = \tilde{\mathbf{A}} \cdot \mathbf{E}_0^{(l-1)}, \;\; \mathbf{E}_0^{(0)} = \mathbf{E}_0 \tag{6}$$

**Efficient and Strong Feature Encoder**. To achieve efficiency while retaining the capacity to encode graph features, our graph experts are configured by deep multi-layer perceptron (MLP) networks. Specifically, the final node embeddings given by an expert model is calculated iteratively as follows:

$$\bar{\mathbf{E}}^{(l+1)} = \mathrm{LayerNorm}\left(\mathrm{Dropout}\left(\mathrm{ReLU}(\bar{\mathbf{E}}^{(l)}\mathbf{W} + \mathbf{b})\right) + \bar{\mathbf{E}}^{(l)}\right) \tag{7}$$

The final embeddings are denoted as $\hat{\mathbf{E}} = \bar{\mathbf{E}}^{(L')} \in \mathbb{R}^{|\mathcal{V}|\times d}$, where $L'$ represents the number of fully-connected layers. And $\bar{\mathbf{E}}^{(0)}$ is initialized by the aforementioned embeddings $\mathbf{E}_1$. Each layer of our MLP module comprises a linear transformation $\mathbf{W} \in \mathbb{R}^{d\times d}$ and bias $\mathbf{b} \in \mathbb{R}^d$, followed by a ReLU non-linear activation, a dropout layer, a residual connection, and layer normalization.

**Multiple Simple Experts as Strong Encoder**. It is worth noting that each graph expert in AnyGraph adopts a very simple learnable network, foregoing the capacity to mine complex hidden relations like those in heavy graph neural networks such as GATs Veličković et al. (2018) and GraphTransformers Hu et al. (2020). This is because AnyGraph employs a MoE architecture, where each expert is expected to handle only a sub-domain of all graph data through simple feature transformations. Therefore, no complex models are needed to accommodate different types of graphs within a single network. Compared to other graph foundation models that rely on a single heavy network, this approach further accelerates the training and inference processes.

## 3.3 EFFICIENT CROSS-DOMAIN MODEL TRAINING

To maximize the cross-graph generalization capabilities of AnyGraph, the training samples from different datasets are mixed together and randomly shuffled during the model training process. Each batch of training samples is composed of the following information:

$$\mathcal{S} = \left(\{(v_{c_b}, v_{p_b})|b \in B\} \subset \mathcal{E}_{\mathcal{G}_s}, \quad \mathbf{E}_1 = \mathrm{InitialEmbed}(\mathcal{G}_s), \quad f_{\Theta_k} \text{ where } k = \psi(\mathcal{G}_s)\right) \tag{8}$$

Inspired by the effectiveness of link-wise graph pre-training tasks Jin et al. (2022), we utilize link prediction as the training task. Here, $(v_{c_b}, v_{p_b})$ denotes the positive edges for link prediction, and $B$ denotes the batch size. To facilitate batch training, each training batch involves only one training graph $\mathcal{G}_s$. The initial node embeddings $\mathbf{E}_1$ and the most competent expert model $f_{\Theta_k}$ are preprocessed in advance to accelerate the training. Specifically, the loss function for AnyGraph is as follows:

$$\mathcal{L} = \sum_{\mathcal{S}} \sum_{b \in B} -\frac{1}{B} \log \frac{\exp(\hat{y}_{c_b, p_b} - \hat{y}_{\max})}{\sum_{v_n \in \mathcal{V}_{\mathcal{G}_s}} \exp(\hat{y}_{c_b, n} - \hat{y}_{\max})} \tag{9}$$

This training objective maximizes the prediction scores for positive samples $(v_{c_b}, v_{p_b})$ and minimizes the predictions for all possible node pairs between $v_{c_b}$ and all nodes $v_n$. To avoid numerical instability, we substract the batch-specific maximum score, $\hat{y}_{\max}$, from all prediction scores.

**Feature and Structure Augmentation**. To enrich training data and enhance input diversity, the training of AnyGraph includes periodic reprocessing of initial graph embeddings $\mathbf{E}_1$ and graph routing results. This reprocessing augments both features and structures, improving AnyGraph's generalizability. • **For initial embeddings**, SVD and simplified GCN processes are periodically reapplied after $|\mathcal{E}|/(10B)$ training steps for each dataset, creating varied embedding spaces and boosting representation heterogeneity. This frequency is adaptive to dataset size to manage computational efficiency. • **For graph routing**, competence scores are recalculated periodically using randomly sampled positive $(v_{c_s}, v_{p_s})$ and negative $v_{n_s}$ pairs. This structural augmentation evaluates graph experts using a random subset, increasing the model's robustness against structural variations.

**Complexity Analysis**. The training and inference process of our AnyGraph involve only a single expert model, yielding a time complexity of $\mathcal{O}(B \times d^2 \times L')$ per batch. Preprocessing of initial embeddings and expert routing does not add to this batch-wise complexity, making AnyGraph significantly more efficient than typical graph foundation models that use complex GNN models such graph transformers. Additionally, expert routing requires $\mathcal{O}\left(\sum_{\mathcal{G}_s} |\mathcal{E}_s| \times d \times K + \sum_{\mathcal{G}_s} |\mathcal{V}_s| \times d^2 \times L' \times K\right)$ computations, with the latter term generally larger and comparable to a simple GCN network. Thus, AnyGraph demonstrates greater efficiency in training and inference compared to existing methods, with the additional routing complexity akin to that of simple GNNs.

## 4 EVALUATION

Our experiments aim to answer the following **R**esearch **Q**uestions:

- **RQ1**: How does the zero-shot predictionperformance of AnyGraph compare to baseline methods?
- **RQ2**: How do AnyGraph's various modules influence its overall performance?
- **RQ3**: How does the model size and the amount of training data impact AnyGraph's performance?
- **RQ4**: How interpretable is the expert routing mechanism within AnyGraph?
- **RQ5**: How is the scalability and efficiency of AnyGraph compared to fine-tuning methods?

### 4.1 EXPERIMENTAL SETTINGS

**Experimental Datasets**. For a comprehensive evaluation of the cross-domain graph generalizability, we employ a total of **38** datasets. These datasets span a wide range of domains, including e-commerce (*e.g.* user interactions and product-wise relations), academic graphs (*e.g.* citation and collaboration networks), biological information networks (*e.g.* relations among drugs and proteins), and other domains like email networks, website networks, trust networks, and road networks.

**Dataset Groups**. We set up different dataset groups and conduct cross-dataset evaluations on these groups. Specifically, all datasets are divided into two cross-domain groups, **Link1** and **Link2**, which have a similar number of total edges and a similar number of domain-specific edges. Additionally, we have three domain-specific groups: **Ecommerce**, **Academic**, and **Others**. The **Others** group is primarily composed of biological networks, combined with other small domains that have fewer datasets. See Appendix A.1 for more information of our experimental datasets.

**Experimental Settings**. We follow previous works (He et al., 2020; Kipf & Welling, 2017) for dataset splitting and evaluation metrics. Our AnyGraph model and the graph foundation models are evaluated on a cross-graph zero-shot prediction task. For baselines that cannot handle cross-dataset transfer, we evaluate their few-shot performance. Details of the evaluation protocols are provided in

Table 1: Comparing AnyGraph (in zero-shot setting) with baseline models (with 5% and 10% training data) on link prediction (Recall@20, NDCG@20), node classification (Accuracy, Macro F1).

| Data | GIN | | | | GAT | | | | GPF | | | | GraphPrompt | | | | GraphCL | | | | **AnyGraph** | |
|---|---|---|---|---|---|---|---|---|---|---|---|---|---|---|---|---|---|---|---|---|---|---|
| | Train 5% | | Train 10% | | Train 5% | | Train 10% | | Tune 5% | | Tune 10% | | Tune 5% | | Tune 10% | | Tune 5% | | Tune 10% | | 0-shot | |
| Metric | R | N | R | N | R | N | R | N | R | N | R | N | R | N | R | N | R | N | R | N | R | N |
| Link1 | 6.46 | 3.06 | 11.80 | 5.45 | 13.52 | 6.65 | 13.45 | 6.78 | 6.04 | 2.92 | 6.80 | 3.27 | 4.33 | 2.24 | 5.42 | 3.11 | 17.23 | 9.00 | 20.55 | 10.76 | **23.94** | **12.68** |
| Link2 | 6.72 | 4.50 | 21.62 | 13.41 | 9.83 | 5.91 | 15.30 | 8.84 | 7.44 | 4.25 | 16.58 | 9.84 | 6.06 | 3.36 | 6.10 | 3.62 | 29.18 | 17.62 | 31.42 | 19.91 | **46.42** | **27.21** |
| Ecom. | 3.36 | 2.58 | 13.41 | 8.06 | 3.79 | 2.94 | 9.64 | 5.78 | 7.25 | 3.84 | 18.72 | 10.94 | 4.90 | 2.59 | 6.06 | 3.36 | 22.13 | 13.19 | 26.05 | 14.59 | **26.92** | **15.05** |
| Acad. | 10.82 | 4.70 | 20.61 | 9.04 | 14.95 | 6.29 | 11.17 | 4.67 | 13.22 | 5.80 | 14.83 | 6.41 | 6.73 | 3.05 | 7.72 | 3.40 | 24.86 | 12.50 | 28.69 | 14.31 | **32.74** | **15.31** |
| Othrs. | 6.92 | 4.46 | 18.43 | 11.85 | 16.34 | 9.22 | 16.17 | 20.88 | 2.40 | 2.12 | 4.51 | 3.44 | 2.93 | 2.36 | 3.42 | 2.72 | 24.54 | 14.93 | 24.62 | 15.90 | **46.83** | **28.97** |
| Metric | Acc | F1 | Acc | F1 | Acc | F1 | Acc | F1 | Acc | F1 | Acc | F1 | Acc | F1 | Acc | F1 | Acc | F1 | Acc | F1 | Acc | F1 |
| Node | 20.79 | 19.46 | 36.04 | 30.60 | 53.76 | 40.14 | 54.83 | 41.61 | 12.77 | 11.45 | 16.29 | 16.00 | 18.01 | 20.59 | 23.15 | 22.89 | 43.70 | 33.72 | 48.75 | 36.15 | **64.31** | **43.24** |

Appendix A.2. The **Hyperparameter Settings** of AnyGraph are provided in Appendix A.3. The compared **Baseline Methods** are introduced in Appendix A.4.

### 4.2 ANYGRAPH'S ZERO-SHOT PREDICTION (RQ1)

To assess the zero-shot performance of AnyGraph, we conducted an extensive evaluation across 38 graph datasets from various domains. We independently trained two versions of the AnyGraph model - one on the Link1 dataset and the other on the Link2 dataset. Each trained model was then used to make zero-shot predictions on the dataset it was not originally trained with. It is important

Table 2: Comparing AnyGraph to existing graph foundation models in zero-shot prediction.

| Method | GraphGPT | | | | OpenGraph | |
|---|---|---|---|---|---|---|
| Data | Pubmed | | Cora | | Ecom. w/o GR | |
| Metric | Acc | MacF1 | Acc | MacF1 | Recall | NDCG |
| Baseline | 0.1813 | 0.1272 | 0.7011 | 0.6491 | 0.1444 | 0.1099 |
| AnyGraph-F | 0.5852 | 0.5325 | 0.7134 | 0.6003 | 0.2281 | **0.1600** |
| AnyGraph | **0.6088** | **0.5492** | **0.7809** | **0.7591** | **0.2382** | 0.1552 |

to note that the Link1 and Link2 datasets do not share the same feature spaces or sources of data collection, which adds to the complexity and challenges of the zero-shot evaluation. The outcomes of this evaluation are detailed in Table 1 and Table 2, and our key observations are as follows:

**i) Superior Generalizability across Diverse Datasets.** • **Superior Prediction Accuracy**. Compared to the few-shot capabilities of existing GNN models, pre-training techniques, and foundation models, AnyGraph demonstrates exceptional zero-shot prediction accuracy across various domains. This superior performance spans both link prediction and node classification tasks. • **Effectively Handling Heterogeneity**. The enhanced generalizability can be attributed to the effective handling of structure-level and feature-level data heterogeneity through unified structure and feature representations in the expert models. This approach enables AnyGraph to develop comprehensive modeling functions that are universally applicable across different graph data scenarios. • **Comprehensive Training**. Additionally, the extensive training regimen, which incorporates a variety of large-scale datasets, equips AnyGraph with a deep and broad expertise in graph learning.

**ii) Limitation of existing pre-training GNNs.** • **Challenges of Cross-Domain Transfer**. Existing pre-training and tuning methods, like GPF, GraphPrompt, and GraphCL, employ self-supervised learning and are pre-trained on half the datasets, then fine-tuned on the remaining datasets using few-shot data. However, this pre-training often fails to yield significant improvements due to substantial distribution disparities across data domains. For instance, datasets may exhibit vastly different link densities or utilize distinct node features, which significantly challenges the transfer of useful knowledge from divergent pre-training datasets during fine-tuning and prediction. • **AnyGraph's Robust Adaptability** To address this challenge, the AnyGraph model incorporates multiple graph expert models tailored to various sub-domains of graph data. This MoE architecture effectively manages datasets from distinctly different domains, such as e-commerce user behaviors, academic networks, and road networks, demonstrating its robust adaptability.

### 4.3 SCALING LAW OF ANYGRAPH FRAMEWORK (RQ2)

In this section, we explore the applicability of the scaling law to AnyGraph. We conduct experiments using 18 different versions of AnyGraph, each differing in model size and quantity of training data. Specific configurations of these variants are discussed in Appendix A.5. The evaluation results are depicted in Figure 3, which includes overall and domain-specific performance, as well as zero-shot and full-shot outcomes. Our key findings are as follows:

**i) Generalizability of AnyGraph Follows the Scaling Law**. As the model size and the volume of training data increase, we notice a saturation point in AnyGraph's full-shot performance. In contrast,

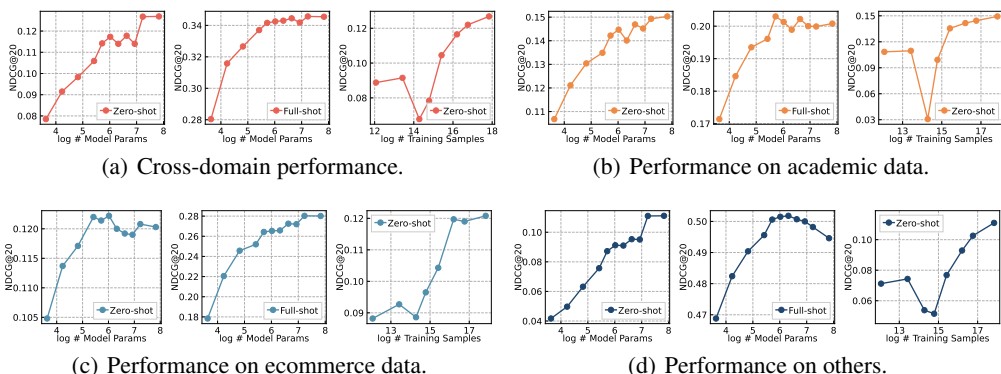

Figure 3: Zero-shot and full-shot performance *w.r.t.* the amount of parameters and training samples.

the zero-shot prediction accuracy continues to improve. This pattern supports the scaling law of graph foundation models, illustrating that scaling up can significantly enhance the capabilities of graph models. Two key factors contribute to this phenomenon:

- **Task Difficulty**. The saturation in full-shot performance is partly because the evaluation tasks might not be challenging enough. In-domain generalization can be more straightforward, leading to a plateau in performance improvements. This insight into the scaling law for graph data encourages further exploration of larger models on more complex graph learning tasks.

- **MoE Architecture**. The integration of the Mixture of Experts (MoE) architecture allows AnyGraph to effectively manage and utilize a broader spectrum of knowledge, particularly in this zero-shot scenario characterized by significant distribution disparities.

**ii) Emergent Abilities of AnyGraph**. The overall zero-shot performance curve illustrates that as the model size increases, the performance sometimes experiences periodic stagnation. With further increments in parameters, AnyGraph's performance undergoes a sudden significant improvement. This phenomenon indicates the emergent abilities of AnyGraph, demonstrating the effectiveness of scaling up in enhancing its generalization capabilities.

**iii) Insufficient training data may bring bias**. In the initial stages of increasing the training data, the introduction of new datasets might negatively impact performance due to their differences from the test graphs. However, this issue can be mitigated by further expanding the training data. By providing the model with a more comprehensive set of training samples, it helps prevent overfitting and reduces bias stemming from dataset disparities.

### 4.4 ABLATION STUDY (RQ3)

This section evaluates the effectiveness of AnyGraph's sub-modules by comparing ablated variants in terms of their zero-shot and full-shot performance across both cross-domain datasets and domain-specific datasets. The results are in Figure 4. We make the following observations:

- **MoE Significantly Enhances Zero-Shot Performance**. The **-MoE** variant, which employs a single expert model without the MoE architecture, demonstrates decent performance on datasets on which it was trained, as shown in parts (b) and (c). However, this variant exhibits a substantial decline in zero-shot prediction capabilities. This underscores the critical role of the MoE architecture in enhancing AnyGraph's generalization abilities. The use of multiple expert models significantly expands AnyGraph's modeling capacity, effectively managing the large disparities between various domains using multiple seperated models.

- **Feature Modeling is Crucial in AnyGraph**. In the -Feat variant, node features are omitted, leading to the most significant degradation in both zero-shot and full-shot performance. This underscores the effectiveness of AnyGraph's unified structure and feature representation method in successfully learning features. This component is crucial for tackling in-domain graph data heterogeneity. Additionally, this outcome highlights the feasibility of unifying different feature spaces created by various methods into a single model for general use.

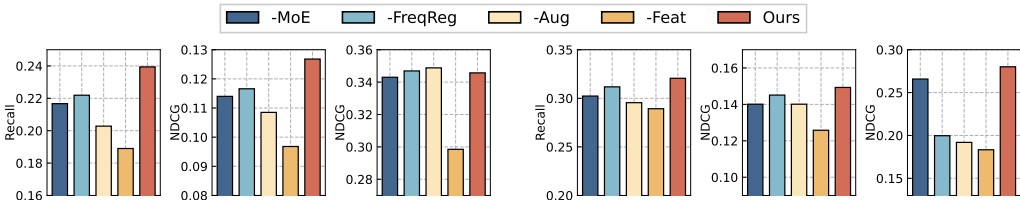

(a) Zero-shot (left & middle) and full-shot (right) performance tested across multiple domains.

(b) Zero-shot (left & middle) and full-shot (right) performance tested on Academic datasets only.

Figure 4: Impact of AnyGraph's sub-modules on zero-shot and full-shot prediction capabilities.

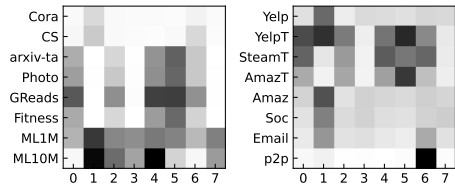

Figure 5: Matching scores between datasets and experts, given by the routing mechanism.

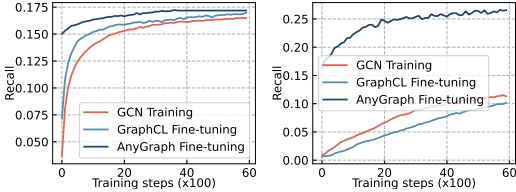

Figure 6: Performance v.s. training/tuning steps, on Citation-2019 (left) and PPA (right) datasets.

- **Effectiveness of Frequency Regularization and Graph Augmentation**. In the **-FreqReg** and **-Aug** variants, the routing adjustment based on the training frequency of experts and the feature and structure augmentation are individually removed. The outcomes from these modifications affirm the beneficial impact of these two components within AnyGraph. Omitting them can lead to biased model training, which undermines the robustness of AnyGraphin handling diverse datasets.

### 4.5 INVESTIGATION ON EXPERT ROUTING (RQ4)

This section delves into the expert routing mechanism of AnyGraph. Figure 5 displays the competence scores of various expert models for the input datasets, as determined by AnyGraph's routing algorithm based on self-supervised loss. The figure illustrates that datasets sharing common characteristics—such as source of collection or feature construction method—are often routed to the same expert models by AnyGraph. For instance, datasets like arxiv-ta, Photo, GoodReads, and Fitness, which utilize a common text-embedding-based feature space, are assigned to highly similar experts. Additionally, ML1M and ML10M, both sourced from the movie-rating platform Movielens, are predominantly associated with expert 1. It is also notable that this routing pattern extends to zero-shot datasets, as shown on the right part of Figure 5. Here, YelpT, SteamT, and AmazonT, which share the same feature space, are assigned to very similar models. This outcome highlights the effectiveness and the explainability of AnyGraph's routing mechanism.

### 4.6 EFFICIENCY STUDY (RQ5)

**Tuning Curve Comparison**. To evaluate the efficiency of AnyGraph, we compare its fine-tuning process with that of GraphCL and the training from scratch process of a GCN model. As depicted in Figure 6, when fine-tuned on a new dataset, the pre-trained AnyGraph rapidly achieves a high performance saturation point. In some instances, such as with the PPA dataset, GraphCL and the end-to-end trained GCN struggle to attain comparable performance levels. This advantage is based on i) the strong cross-domain generalization capabilities of AnyGraph, which bring a high starting point for the new dataset, and ii) the efficiency of AnyGraph's MoE architecture, which requires only one MLP network for efficient but effective modeling and parameter tuning.

In addition, it is observed that pre-training GraphCL does not always benefit its fine-tuning on new datasets, as evidenced by GraphCL's underperformance relative to GCN in Figure 6 (right). This is due to the large distribution gap between the pre-training data Link2 and the test data PPA.

**Training Time Comparison**. To evaluate the efficiency of the models under consideration, we compared the training times of the three models. As indicated in Table 3, AnyGraph, despite having significantly more parameters, has training times that are comparable to, or even less than, the other two models. This underscores the efficiency of our model design.

Specifically, AnyGraph avoids the cumbersome full-graph propagation. Instead, it utilizes structure-aware embeddings derived through a non-trainable pre-processing method. This significantly reduces both the time and memory requirements. Furthermore, the MoE

Table 3: Training time for each 100 steps.

| Dataset | CS | ML1M | Yelp | Email | Cite19 | roadNet | PPA |
|---|---|---|---|---|---|---|---|
| GCN | 1.5s | 4.2s | 6.0s | 2.5s | 19.2s | 27.8s | 101.1s |
| GraphCL | 1.1s | 4.9s | 9.4s | 2.8s | 43.1s | 57.1s | 130.8s |
| Ours | 1.5s | 3.5s | 6.1s | 3.0s | 31.6s | 37.3s | 41.1s |

architecture equips AnyGraph with the capability to use only $1/K$ of the computational resources for most prediction and optimization processes, thereby greatly reducing overall computational costs.

## 5 RELATED WORKS

**Graph Neural Models**. Graph learning has garnered significant interest for its broad applicability across various fields such as user behavior modeling and biology/chemistry applications (Chang et al., 2021; Hao et al., 2020). Graph neural networks (GNNs) learn node representation vectors for downstream tasks like node classification and link prediction. The core mechanism involves iterative message passing, refining node embeddings to capture both node-specific information and higher-order topological structures. This process ensures that the final node embeddings effectively encapsulate both node-specific information and higher-order topological structures. Notable techniques include Graph Convolutional Networks (GCNs) (Jin et al., 2021), Graph Attention Networks (GATs) (Brody et al., 2022), Graph Isomorphism Network (GIN) (Xu et al., 2018), and Graph Transformer (Hu et al., 2020), which improves the graph modeling abilities. Despite the advancements, these methods remain reliable on high-quality training data and often struggle with generalization.

**Self-Supervised Graph Learning**. Given the challenges with the generalizability of GNNs, considerable research efforts (Xie et al., 2022) have focused on enhancing GNNs through self-supervised learning objectives, aiming to capture invariant graph features. Specifically, GraphCL (You et al., 2020) introduced a contrastive pre-training approach for graph data, designed to learn authentic graph characteristics that are robust to structural and feature perturbations. Building on this, JOAO (You et al., 2021) and GCA (Zhu et al., 2021) have developed adaptive augmentation strategies for self-supervised tasks, effectively mitigating the adverse effects of random augmentations. Subsequent works have sought to quickly adapt these pre-trained models to downstream tasks and evolving graph data, as demonstrated by GPF (Fang et al., 2023) and GraphPrompt (Liu et al., 2023). Despite the success, the generalizability of these methods remains confined to graph data with similar structural and feature patterns, overlooking the cross-domain generalization challenge highlighted in our paper.

**Large-scale Graph Pre-training**. Recent advances in graph modeling have seen efforts to pre-train large-scale graph models across multiple datasets to improve their generalizability, drawing inspiration from the strong generalization capabilities of large language models (LLMs). For instance, OFA (Liu et al., 2024) and ZeroG (Li et al., 2024) utilize text embeddings to standardize the feature spaces across various graph datasets and tasks, facilitating cross-dataset training of graph models. Models like InstructGLM (Ye et al., 2024) GraphGPT (Tang et al., 2024a) and LLaGA (Chen et al., 2024) synchronize graph representation spaces with the hidden spaces of LLMs, thus enabling the application of general language models for graph prediction tasks. Furthermore, HiGPT (Tang et al., 2024b) expands the capabilities of LLMs to accommodate heterogeneous graph data.

Despite these advancements, most generalized graph models require substantial access to and integration of text features, which confines their use primarily to text-abundant environments such as academic networks. Additionally, these methods are typically trained within specific application realms, failing to address the significant variances between datasets from diverse domains.

## 6 CONCLUSION

In this work, we present the AnyGraph framework, an effective and efficient graph foundation model designed to address the multifaceted challenges of structure and feature heterogeneity across diverse graph datasets. AnyGraph's innovative Mixture-of-Experts (MoE) architecture, coupled with its dynamic expert routing mechanism, positions it at the state-of-the-art of cross-domain generalization capabilities. Extensive experiments on 38 varied graph datasets have not only underscored Any-Graph's superior zero-shot learning performance but also its robustness to distribution shifts and its adherence to scaling laws, thereby enhancing its predictive accuracy with increased model size and data volume. The model's efficiency in training and inference, validated through comparison with existing methods, further cements its practical applicability.

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

## A  APPENDIX

### A.1  EXPERIMENTAL DATASETS

We utilize a total of 38 graph datasets across various domains. The entire experimental data contains 14,437,372 nodes, and 199,265,688 edges. The dataset specifics are detailed below:

**E-commerce Datasets**. This category includes 15 datasets from various e-commerce contexts such as user rating platforms and online retail services. These datasets vary in terms of the presence and type of node features. For instance, datasets such as Amazon-book, Yelp2018, Gowalla, Yelp-text, Amazon-text, Steam-text, Goodreads, Amazon-Fitness, Amazon-Photo, Movielens-1M, Movielens-10M, Products-home, Products-tech, Home-node, Tech-node are included. Notably, Amazon-text, Steam-text, and Yelp-text utilize the same method for feature generation, while Fitness, Photo, and Goodreads employ a different consistent method.

**Academic Network Datasets**. We use 13 datasets focused on academic networks, which include citation and collaboration relations among scholars and papers. These datasets represent various research fields and employ diverse feature generation methods, such as NLP embeddings, bag-of-words, and different versions of large language models. The specific datasets are Cora, Pubmed, Arxiv, Cora-link, Pubmed-link, Citeseer, CS, Arxiv-link, Arxiv-t (with features derived using an alternative method), Cite-2019, Cite-20Cent, OGB-Collab.

**Biological Information Networks**. Our experimental data includes 6 datasets related to biological entities like proteins, drugs, and diseases. This category features networks such as OGB-DDI, OGB-PPA, which record drug-drug and protein-protein relations, respectively, and four other protein relation networks for different species, denoted as Proteins-0, Proteins-1, Proteins-2, Proteins-3.

**Other Datasets**. In addition to the categories mentioned above, we include 5 datasets from various other fields: an email network Enron, a website network Stanford, a road network dataset Road-PA, a P2P web network dataset Gnutella, and a trust network dataset Epinions.

**Dataset Groups**. For conveinience of performance evaluation, we split the many datasets using different grouping methods. Firstly, two big data groups Link1 and Link2 are made using all the link prediction datasets. Notably, datasets from the same source of collection, such as ML-1M and ML-10M, or uses the same method to generate features, such as Fitness, and Photo, are put into the same group, to avoid information leakage when evaluating zero-shot performance on the other group. Apart from these two datasets, we also conduct evaluations on domain-specific groups, including E-commerce, Acadmic, and Others. Specifically, these data groups contain the following datasets:

- **Link1**: Products-tech, Yelp2018, Yelp-text, Products-home, Steam-text, Amazon-text, Amazon-book, Cite-2019, Cite-20Cent, Pubmed-link, Citeseer, OGB-PPA, Gnutella, Epinions, Enron.

- **Link2**: Photo, Goodreads, Fitness, Movielens-1M, Movielens10M, Gowalla, Arxiv, Arxiv-t, Cora, CS, OGB-Collab, Proteins-0, Proteins-1, Proteins-2, Proteins-3, OGB-DDI, Stanford, Road-PA.

- **Ecommerce** and **Academic**: These groups contain all domain-specific datasets mentioned above.

- **Others**: This group contains all the biological datasets mentioned above, and datasets from other minor domains, including email network data Enron, website network data Stanford, road network data RroadNet-PA, P2P network data Gnutella, and trust network data Epinions.

### A.2  EVALUATION PROTOCOLS

All datasets used in this study are sourced from previous research as referenced (Tang et al., 2024a; Li et al., 2024). We adhere to the original data splits from these sources to delineate our training and testing sets. Given that many baseline methods are not equipped to manage zero-shot prediction across datasets, we instead assess their few-shot capabilities. This allows for a comparative analysis against the zero-shot performance of AnyGraph. We employ specific evaluation settings tailored to each method, detailed as follows:

- **Zero-shot Setting for AnyGraph, GraphGPT, and OpenGraph**. In our study, AnyGraph and two comparative graph foundation models, GraphGPT and OpenGraph, undergo evaluations for zero-shot prediction capabilities. We pre-train two instances of AnyGraph using Link1 and Link2

Table 4: Statistics of the experimental datasets.

| Dataset | DDI | Collab | ML1m | ML10m | Amazon-book | PPA | Yelp2018 | Gowalla | Cora | Pubmed | Citeseer |
|---|---|---|---|---|---|---|---|---|---|---|---|
| # Nodes | 4,267 | 235,868 | 9,746 | 80,555 | 144,242 | 576,289 | 69,716 | 70,839 | 2,708 | 19,717 | 3,327 |
| # Edges | 1,334,889 | 1,285,465 | 920,193 | 9,200,050 | 2,984,108 | 45,495,642 | 1,561,406 | 1,027,370 | 10,556 | 88,648 | 9,104 |
| $d$ Feats | 0 | 128 | 0 | 0 | 0 | 58 | 0 | 0 | 1433 | 500 | 3703 |

| Datasets | Proteins-0 | Proteins-1 | Proteins-2 | Proteins-3 | Products-home | Products-tech | Yelp-t | Amazon-t | Steam-t | Goodreads | Fitness |
|---|---|---|---|---|---|---|---|---|---|---|---|
| # Nodes | 25,449 | 6,568 | 18,108 | 13,015 | 9,790 | 47,428 | 22,101 | 20,332 | 28,547 | 676,084 | 173,055 |
| # Edges | 11,660,646 | 1,845,960 | 7,418,688 | 3,962,930 | 131,843 | 2,077,241 | 277,535 | 200,860 | 525,922 | 8,582,306 | 1,773,500 |
| $d$ Feats | 0 | 0 | 0 | 0 | 100 | 100 | 1536 | 1536 | 1536 | 768 | 768 |

| Datasets | Epinions | Enron | Stanford | Road-PA | Gnutella | Cite-2019 | Cite-20Cent | Arxiv | Arxiv-t | Photo | CS |
|---|---|---|---|---|---|---|---|---|---|---|---|
| # Nodes | 75,879 | 36,692 | 281,903 | 1,088,092 | 8,717 | 765,658 | 1,016,241 | 169,343 | 169343 | 48,362 | 18,333 |
| # Edges | 508,837 | 183,831 | 2,312,497 | 1,541,898 | 31,525 | 1,917,381 | 5,565,798 | 1,166,243 | 1,166,243 | 500,939 | 163,788 |
| $d$ Fets | 0 | 0 | 0 | 0 | 128 | 128 | 128 | 128 | 768 | 768 | 6805 |

datasets. The model pre-trained on Link1 is then tested for zero-shot performance on the Link2 group datasets, and vice versa. Results labeled as "zero-shot" for AnyGraph are derived using this cross-evaluation method. Conversely, results marked as "full-shot" pertain to supervised learning outcomes, where, for example, the model trained on Link1 is tested on the test sets of Link1 group datasets. For GraphGPT and OpenGraph, we utilize the models as released in their respective original studies, which were pre-trained on specified datasets.

- **Zero-shot Node Classification for AnyGraph**. Inspired by prior research (Sun et al., 2022), we approach zero-shot node classification by representing node classes as distinct nodes. We then connect existing nodes that have training labels directly to these new class nodes. This technique eliminates the need for learning specific parameters for each class within the zero-shot learning framework, streamlining the process. We have integrated this innovative approach into baseline methods as well, enhancing their capability to handle unseen node labels effectively.

- **Few-shot Training for GIN and GAT**. The GIN and GAT models, employed as end-to-end training baselines, undergo training from scratch on few-shot subsets of the evaluation datasets. This approach is necessary because these models are not well-suited for cross-dataset transfer, particularly when dealing with datasets that have varying feature dimensionalities.

- **Pre-training and Few-shot Tuning for GraphCL, GPF and GraphPrompt**. These category of baselien methods follow the pre-training-and-fine-tuning mode. In our evaluations, they are firstly pre-trained using the same pre-training datasets as our AnyGraph. Then, they experience an additional fine-tuning process using the few-shot subsets of the evaluation datasets.

**Evaluation Metrics**. For link prediction, we follow previous works (He et al., 2020) and utilize Recall@20 and NDCG@20 as the evaluation metrics. Note that we typically use the summary results of the evaluation results across multiple datasets. Results for fifferent datasets are averaged according to their number of test samples. For the node classification task, we employ the widely-used Accuracy and Macro-F1 score as our metrics (Chen et al., 2022; Tang et al., 2024a).

## A.3 HYPERPARAMETER SETTINGS

**Optimization**. Our model, AnyGraph, is implemented using PyTorch. The optimization process employs the Adam optimizer with a learning rate of $1 \times 10^{-4}$ and a training batch size of 4096. We use cross-entropy loss with a sampled negative set (Wu et al., 2021). The learnable parameters of AnyGraph are initialized using the Xavier uniform initializer. **Network Configurations**. The standard configuration of our AnyGraph includes 512 hidden units and 8 graph expert models. Each expert model comprises 8 fully-connected layers. These layers utilize a ReLU activation function and incorporate a dropout layer with a dropout probability of 0.1. **Algorithm Hyperparameters**. The frequency regularization of our routing mechanism is set with an adjustment range of $\rho = 0.2$. The SVD decomposition is performed using 2 iterations. For structural and feature augmentation, each dataset is reprojected after using 1/10 of its samples for optimization. A minimum of 100 training steps should be executed for each dataset before its initial representations are reprojected. The reassignment of experts occurs after all training datasets have undergone one cycle of re-projection.

The baseline methods are evaluated using theeir original code or released model. We closely follow the original code to adapt to our experiments. Grid search is conducted to search for the best hyperparameter settings for each baseline method.

## A.4 BASELINE METHODS

This section provides detailed descriptions of the baseline models used in our analysis. We employ seven different baseline models across four distinct categories.

**Training-from-scratch Graph Neural Networks**.

- **GAT** (Veličković et al., 2018). Graph Attention Networks (GAT) leverage an attention mechanism to dynamically weight node-to-node connections, enhancing the model's ability to adaptively propagate and aggregate information across the graph.

- **GIN** (Xu et al., 2018). The Graph Isomorphism Network (GIN) significantly boosts the expressive power of Graph Neural Networks by introducing a unique graph encoding technique aimed at effectively distinguishing between non-isomorphic graphs.

**Graph Pre-training Models**.

- **GraphCL** (Zhu et al., 2021). It enhances the pre-training of graph models via self-discriminative contrastive learning, which is applied to learned node embeddings. The method employs various graph augmentation techniques such as node dropping, edge permutation, random walks, and feature masking to improve robustness.

**Graph Prompt Tuning Methods**.

- **GraphPrompt** (Liu et al., 2023). It proposes a unified approach that integrates pre-training and prompt tuning for graph models. It features a learnable prompt layer designed to automatically extract crucial information from the pre-trained model to enhance downstream performance.

- **GPF** (Fang et al., 2023). The Graph Prompt Framework (GPF) is a versatile graph prompt tuning framework compatible with various graph pre-training methods. It offers two variants of a learnable graph prompt layer, tailored to different application needs.

**Graph Foundation Models**.

- **GraphGPT** (Tang et al., 2024a). This approach proposes representation alignment and instruction tuning techniques to align graph representation spaces with text encoding spaces, empowring large language models with the capabilities of zero-shot graph encoding and inference.

- **OpenGraph** (Xia et al., 2024). This method introduces a unified graph tokenizer, a scalable graph transformer to improve the model's performance and generalization ability. An LLM-enhanced data augmentation mechanism is proposed to address domain-specific data scarcity.

## A.5 DETAILS OF THE SCALING LAW EXPERIMENT

For the scaling law experiment (RQ2), we elaborate the configurations of the developed instances of AnyGraph. For AnyGraph with different model sizes, we begin with the smallest model which has 64 hidden units, 1 fully-connected layer, and 1 expert model. The subsequent 3 model instances increases in their hidden dimensionality, from 64 to 128, 256, and 512. Then 3 larger models with more fully-connected layers are utilized, respectively containing 2, 4, and 8 MLP layers. Then we have MoE versions of AnyGraph, with 2, 4, and 8 experts, respectively. The final largest instance of AnyGraph has a larger latent dimensionality of 1024.

For the increase of training data, we begin with a subset of Link2 data including Cora and CS. The next version additionally includes Photo. The thir one includes ML1M. The fourth one includes Gowalla. The fifth one additionally include Arxiv and Arxiv-t. The sixth one adds the following datasets: collab, ddi, Yelp2018, Fitness, proteins-spec1, web-Stanford, proteins-spec3. The seventh one is trained with proteins-2, roadNet-PA, and Fitness additionally. And the final one is trained with all datasets from Link2. In this manner, we gradually increase the amount of training data.

## A.6 SUPPLEMENTARY EXPERIMENTAL RESULTS

**Model Performance Curves**. We monitored the training loss and test performance of AnyGraph across each training epoch to understand its training dynamics. This included evaluating AnyGraph's performance on the test sets of its training datasets (full-shot performance) as well as its performance on unseen datasets (zero-shot performance), as depicted in Figure 7.

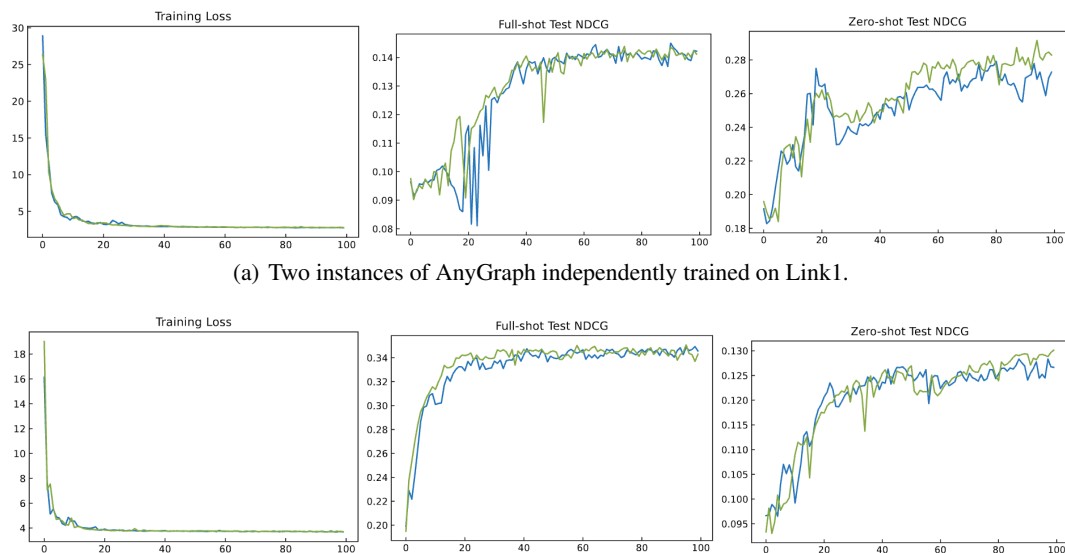

(a) Two instances of AnyGraph independently trained on Link1.

(b) Two instances of AnyGraph independently trained on Link2.

Figure 7: Training loss, test NDCG of full-shot and zero-shot prediction, v.s. the number of training epochs. Two curves in each plot correspond to two independently-trained instances of AnyGraph.

The analysis reveals that training loss and full-shot test performance stop to decrease/increase significantly after approximately 40 epochs. In contrast, zero-shot test performance continues to improve significantly, even up to 100 epochs. This trend underscores a steady enhancement in the model's generalization abilities, highlighting the potential to further explore and enhance the generalizability of graph models in challenging zero-shot inference tasks.

**Performance on Industrial Data**. We further assessed the performance of AnyGraph using a real-world dataset from a popular user reading platform, comprising over 1 million user and item nodes. We trained a base graph neural model on historical user behavior data, and evaluated both the base model and

Table 5: Performance on industrial data.

| Method | History | 10% | 20% | 30% | 40% | 50% |
|---|---|---|---|---|---|---|
| Base Method | 0.7% | 2.0% | 5.6% | 10.6% | 17.3% | 19.9% |
| AnyGraph | 6.3% | 3.4% | 7.5% | 14.0% | 19.3% | 21.7% |

AnyGraph using varying amounts of new interaction data to construct the input graph. The results, summarized in Table 5, show that "History" indicates the base model was trained on data from previous days, while "10%", "20%", etc. represent the percentages of new data used to construct the input graph. Importantly, the new data was used only as input features, not for tuning, reflecting a real-world scenario where models cannot be promptly fine-tuned on new data. Our key observations are: i) AnyGraph demonstrated superior zero-shot predictive capabilities, outperforming the base model trained on historical data. ii) This underscores the importance of robust zero-shot prediction, as new data may not align with historical patterns in real-world settings.

**Recall@20 for Full-shot Performance in Ablation Study**. We have expanded our analysis to include full-shot prediction performance, as assessed in our ablation studies. Figure 8 displays the performance of various ablated versions of our AnyGraph alongside the complete model, using Recall@20 as the metric. A notable finding, absent from the original results, is that removing the augmentation actually results in a significant advantage for our AnyGraph in cross-domain evaluations. This phenomenon can be attributed to the fact that data augmentations interfere

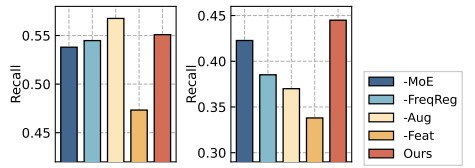

Figure 8: Recall results of ablation study, on cross-domain (left) and academic (right) data.

with the optimization of AnyGraph on the training dataset, thereby impairing the full-shot performance on seen datasets. However, as the zero-shot performance test results indicate, this augmentation technique substantially enhances the generalization capability of AnyGraph. This is because the disturbances prevent the model parameters from overfitting to the training data.

