# OpenReview forum: "AnyGraph: Graph Foundation Model in the Wild"
_ICLR.cc/2025/Conference — ICLR 2025 Conference Withdrawn Submission_

### Official Review · Reviewer_7fyJ · 2024-10-15

**Soundness:** 2
**Presentation:** 3
**Contribution:** 2
**Rating:** 3
**Confidence:** 4

**Summary:**

This paper propose graph MoE with some training strategies for a graph foundatoin model.

**Strengths:**

S1: Nice presentation.

S2: their target problem is very important.

S3: The ambition of exploring a graph foundation model, especially getting rid of LLMs, should be encouraged, which is rare and commendable.

**Weaknesses:**

W1: My 1st concern is about the semantic unifying. Different graphs usually have totally different features w.r.t semantics and dimensionality. The solution of this paper is to use SVD for dimension alignment and link prediction as a training target. However, it should be noted that even if you use SVD to achieve the same dimensionality, they are still located in different latent semantic spaces. The natural gap in the semantic space will not be narrowed with SVD tricks, and the link prediction is not helpful for further semantic alignment.

I personally hold the following opinion (I am open to seeing further discussion if my opinion is not accepted by the authors or other reviewers): currently, the general feature alignment issues across different graph datasets is still an irresolvable problem, unless we have some detailed assumption (e.g. two graph datasets share some overlapped instances, anchor nodes, etc.)

With the above discussion, I think the reason why this paper works w.r.t feature gap (Feature Heterogeneity as they called) may be from some good luck. For example, the experimental datasets might in nature share a large body of their feature semantics (making SVD spaces also overlap) but this is not related to their technique contributions.


W2: The 2nd concern is about structure heterogeneity. I didn't find the corresponding solution to this problem from their whole framework. In particular, the gold standard to convince the readers that you solve the structure heterogeneity issue is the combination of both homophilic and heterophilic graph datasets. The authors should see whether it will cause negative transfer issues, or positive performance than a single model. Currently, the evaluated datasets seem to be all homophilic datasets.

W3: The 3rd concern is about the truth of the Scaling Law of AnyGraph. It is interesting to see some possible trends of scaling law from Figure 1 and Figure 3. However, I personally hold a very cautious opinion of this trend, and I think we need more convincing evidence.  My concern comes from a very natural comparison: language texts only share several thousands of words (a.k.a tokens) but they collect an enormous corpus to train a very large language mode and then they see the scaling law phenomenon. Compared with text as a linear structure with limited tokens, graph datasets are a non-linear structure with nearly unlimited tokens (nodes), but the training graph dataset scale and the parameters of the existing framework are far from comparable to their counterparts in NLP. This is far-fetched to convince the readers that the observation is indeed Scaling Law. For example, how about extending the x-axis further? Currently, the best performance is still not significantly higher than the lowest performance (they seem to be all around 0.1-0.2).

W3: I feel confused about the claim like cross-domain and cross-dataset. According to their experimental settings, link 1 and link 2 have some overlap domains so the results of such a training strategy might not be appropriate to be claimed as cross-domain effectiveness.

W4: I did not find something new in technical contributions. I neither find something huge in model size and data size like their counterpart in the NLP area. I think either of these two aspects is a notable contribution if it indeed works.


W5: Currently, the graph foundation model might be more promising to achieve within a specific domain rather than general multiple domains. I think their claim is a little exaggerated and over-claimed. Compared with the foundation model in NLP, I think the graph foundation model is far from feasible at the current stage. We should first solve some very fundamental problems. for example, how to solve unlimited tokens in graph datasets (while in the NLP area, tokens are limited). How to increase the complexity of the graph model and make sure they will not undergo performance bottleneck (e.g. oversmooth issues for traditional GNNs). How to build a very large dataset and make sure the graph foundation model can learn so-called "knowledge", and furthermore, whether the concept "knowledge" truly exists in the graph domain compared with their counterpart in the NLP area.  Currently, the academic can even not come to an agreement on graph knowledge learned by existing pre-training approaches. I am open to seeing some revolutionary breakthrough, some entirely new infrastructure to replace existing graph models and then I think we might have some brittle confidence to modestly say that we might come closer to the "graph foundation model".

W6: There is a very weird thing: The authors compare their so-called "graph foundation model" with graph-promoting methods. I believe the authors might be lost in these concepts. Graph prompting is used to prompt a pre-trained graph model for more general task performance. Currently, studies on graph prompts face a very awkward situation: compared with their counterpart in LLMs, we do not have a very large graph foundation model. So the researchers on graph promoting have no choice but to deploy their graph prompting technique on GNNs. However, since the authors claim that they successfully invented the "graph foundation model", there is no doubt that we should immediately apply graph prompting techniques to your graph foundation model to see what will happen. But the weird thing is that you compare your graph foundation model with graph prompts. It is a logical mess. It is just like, inventing GPT 4.0, and then comparing your GPT 4.0 with GPT 1.0 with a prompt and then claiming that prompt is not better than GPT 4.0. I do not know what the points of this comparison are. One is the model, one is the usage process. I feel confused about the comparisons and the results in Table 1 (w.r.t graph prompt) make no sense.

**Questions:**

see W1-W6

---

### Official Review · Reviewer_QRDw · 2024-10-24

**Soundness:** 2
**Presentation:** 1
**Contribution:** 2
**Rating:** 5
**Confidence:** 4

**Summary:**

The paper proposes the use of k expert for classification purposes. Each expert model is trained on edge samples (observed and unobserved edges) from multiple graphs (graphs are assigned with a problem routing algorithm) using a link prediction approach. Each expert model generates a |V|x d-dimensional embedded matrix by combining the SVD of the Laplacian-normalized adjacency matrix and the SVD of the node feature matrix (equation 5). This embedded matrix is modified and a multilayer perceptron (MLP) is applied for further modifications generating the final embedding (the training process of the MLP is based on the link prediction problem). When a new graph arrives, a competence score is calculated per expert model (equation 4). The score combines the embedding of each expert and the probability of predicting a sample (observed and unobserved edges) of the target network. The expert with the highest score is used for the final classification problem.

**Strengths:**

The paper proposes a complex framework, that seems to work, but details are not enough to evaluate properly (even if I consider the appendix).

The experiment section has several analyses making the paper stronger. However, important details are omitted.

**Weaknesses:**

Please, rewrite the abstract. According to the current abstract, the main contribution is a unified graph model, but the problem is not described. Is this unified model to generate a graph, to learn the distribution of the graph, or any other problem? Something similar happens with the introduction, where the characteristics of the model are mentioned, but not the problem. Based on page 3 (line 111), it can be deduced that the model is for prediction. Finally, in section 2, preliminaries, the problem is mentioned/defined in more detail.

The readability of the paper is very low. Besides the issue of the problem definition explained before, the methodology also lacks several details, forcing the reader to derive these details. For example, in equation 3, the value S is not defined, and it must be deduced that S is the number of samples used for the graph. Also, given the importance of \hat{\mathbf{e}}_i, be more specific, and mention that \hat{\mathbf{e}}_i is the node embedding of v_i. How do you sample the nodes? What is the effect of the sampling process on the competence indicator? What is the effect of the encoder on the competence indicator function (no embedding process is mentioned until the next subsections)?

The notation could be improved. Equation 3 has a subindex k which corresponds to the value of the k-expert, but as can be appreciated, the value k is not in the equation. Again, the reader must deduce that the embedding is specific for each expert and that the embedding used in that equation corresponds to the embedding of the expert k. A simple solution is to include another subscript/superscript making a refer to k.

According to the Training Frequency Regularization, it seems that the graph expert routing mechanism is used to assign a graph to an expert in the training process ("These models generally receive more or better training samples in the early training stages, giving them an advantage over other experts"). However, it is mentioned that each expert will be focused on graphs with specific characteristics. How do you ensure that each expert can model the different characteristics, even considering that the sample of the graph can not be representative?

Be careful with the statement "This approach consumes only 1/K of the computational and memory resources required for predictions and optimization, compared to other non-MoE". You should prove this with a memory complexity analysis. Consider the same model training one time with all the data, against K equal models each of them trained with different data. In both cases, the estimation will be the same; however, the proposed method will have to analyze K competence indicator increasing the time.

The paper is not self-contained. Most details for reproducibility are in the appendix instead of the main paper. Recall, we are not forced to read the Appendix. What is your first experiment? Did you randomly choose some edges or did you try all possible combinations of edges and use the top 20 edges for prediction? Why the Table 1 changes the Data for the different models?

What do you mean by a zero-shot prediction task? Zero-shot usually means without any further training process. In the case that you are using the same data used for a previous training approach, then this is not a zero-shot prediction task. In the case that you are using another dataset, then you are analyzing the generalization capability of the model, which could be interesting. However, given the lack of details of the experiment section and the appendix, this can not be concluded.

Why do you think that the model is able to classify correctly the node labels even though they are not even considered in the process?

How did you setup the other baseline models? Did you use the same datasets as the data used in the proposed framework?  Why you did not use other models for Table 2? What is AnyGraph-F?

In summary, the paper seems to be an ensemble method, which is described as a complex framework, and important details for the evaluation process are not mentioned.

Minor comments:
Line 94 "topologies.Furthermore"
Figure 1 is not mentioned in the text.
Line 127 d_0 is not defined
Line 128 "indicating"
Line 242 Acronyms must be defined before their are used

**Questions:**

Please see the questions written in the previous section.

---

### Official Review · Reviewer_C6C2 · 2024-10-26

**Soundness:** 3
**Presentation:** 3
**Contribution:** 3
**Rating:** 5
**Confidence:** 4

**Summary:**

This work tackles an important problem for graph learning methods, which is to equip graph learning models with the capability to be able to efficiently adapt to a wide range of graph domains and tasks. To that end, this paper proposes AnyGraph, which aims to address several challenges for graph foundation models, such as dealing with structure and feature heterogeneity of real-world graphs, and the ability to quickly adapt to new domains, and follow the scaling law. AnyGraph achieves this goal by using several interesting ideas, including graph mixture-of-experts and the use of a unified mapping function for graphs and features. Extensive experiments on 38 graph datasets demonstrate the effectiveness of AnyGraph, such as its superior zero-shot prediction performance compared to the few-shot performance of several baselines.

**Strengths:**

S1. This work tackles an important problem for graph learning methods, which is to equip graph learning models with the capability to be able to efficiently adapt to a wide range of graph domains and tasks. Due to the diverse and heterogeneous nature of graph datasets (e.g., graph-structural heterogeneity and feature heterogeneity), existing approaches yet struggle to satisfy several desiderata for dealing with such real-world data, such as fast adaptation to new datasets, broad applicability, and adherence to the scaling law. This paper fills in this gap by developing a new approach that tackles these challenges.

S2. This paper proposes AnyGraph, a new graph foundation model that tackles the aforementioned challenges using several interesting ideas, including graph mixture-of-experts, graph expert routing, and the use of a unified mapping function to process heterogeneous graph connectivity patterns and node features. The proposed model architecture and expert routing mechanism enable an efficient training and inference, while achieving high generalization performance.

S3. The paper presents extensive experimental results using 38 graph datasets, which shows the effectiveness of AnyGraph, such as its superior zero-shot prediction performance and efficiency, and demonstrates that the performance of AnyGraph follows the scaling law. In comparison to the baselines, which were trained using 10% of the training data, the proposed AnyGraph’s zero-shot performance is much better, which is an impressive result. Also, the ablation studies show the positive contributions the individual components of AnyGraph make to the overall zero-shot performance.

**Weaknesses:**

W1. While the most important capability of AnyGraph would be the ability to effectively generalize to new domains, it is not clear how the design of AnyGraph, in principle, can achieve strong generalization capability when given graphs from new domains. Despite strong experimental results involving cross-domain groups, it is unclear how AnyGraph can generalize to new domains that are significantly different from the domains observed during training. Given a graph from a new domain, AnyGraph will select an expert that has the best competence score. However, that doesn’t mean that the selected expert could perform well, if the domains the selected expert has been trained on significantly differ from the new domain. Thus, when given a graph that greatly deviates from observed graphs, it doesn’t seem like any trained expert could perform well in principle.

W2. The Related Works section lacks a discussion of mixture-of-expert models and graph foundation models. While graph mixture-of-experts lies at the center of the proposed AnyGraph’s architecture, it is not discussed how the proposed graph mixture-of-experts framework relates to, and/or improves upon, previous (graph) mixture-of-expert models, which makes it hard to judge the technical contributions of the proposed framework. Also, a discussion of graph foundation models in the Related Works section would be helpful for a clearer contextualization of AnyGraph.

W3. Some of the design choices of AnyGraph seem a bit arbitrary, and need better justifications, despite its strong empirical performance. For instance, in Eqn (5), the Flip() function aligns important features of A with less important features of F. This alignment seems counter intuitive, and it is not clear how this helps. What is the intuition behind this? Also, in Eqn(4) for training frequency regularization, what is the intuition of adding +1.0-\rho/2?

W4. It is not clear how AnyGraph compares with baselines in terms of number of parameters. While the paper says that AnyGraph has significantly more parameters than baselines (Sec 4.6), it’s not clear how this can be the case. AnyGraph does not seem to have many parameters due to its use of simplified GCN. Given that the model size (# params) is one of the major axes that constitutes the scaling law, it would be useful to provide a comparison among representative methods in terms of the number of parameters.

**Questions:**

Q1. In the Feature and Structure Augmentation paragraph, how can feature reprocessing create varied embedding spaces? In my understanding, the initial embeddings should remain the same since no learnable parameters are involved with this process.

Q2. In Table 1, what datasets were used for training and evaluation for node classification tasks?

Q3. Table 2 is confusing to read. There are columns for GraphGPT and OpenGraph, and rows for Baseline, AnyGraph-F, and AnyGraph. What does each entry mean? What does Baseline refer to? What is AnyGraph-F?

Q4. Dataset sources should be specified with the corresponding citations. No citations are currently given in Appendix A.1.

---

### Official Review · Reviewer_Kroa · 2024-11-01

**Soundness:** 2
**Presentation:** 1
**Contribution:** 2
**Rating:** 3
**Confidence:** 4

**Summary:**

Graph foundation models offer a transformative solution, with the potential to learn robust, generalizable representations from various graph data. This paper studies the key challenges of building a graph foundation model: 1/ structure heterogeneity of different graphs, 2/ feature heterogeneity of different graphs, 3/ foundation model adaptation to new domains. The paper proposes a unified graph model AnyGraph to handle the challenges. AnyGraph uses singular value decomposition (SVD) to extract graph structure information and feature information to address the in-domain graph heterogeneity. AnyGraph employs a MoE architecture consisting of multiple graph expert models to address the cross-domain graph heterogeneity. The evaluation results show that the proposed graph foundation model AnyGraph outperforms the baseline GNN models like GIN, GAT, etc, on the zero-shot/few-shot setting.

**Strengths:**

* The paper highlights the key challenges of building a graph foundation model: 1/ structure heterogeneity of different graphs, 2/ feature heterogeneity of different graphs, 3/ foundation model adaptation to new domains. It is a good summary of existing challenges.

* AnyGraph employs a MoE architecture to tackle the cross-domain graph heterogeneity. It proposes a AnyGraph specific method to route the experts. Specifically, the expert indicator score is computed with dot-product-based relatedness scores for positive and negative edges of an input graph. The method is soundness and effective.

**Weaknesses:**

* The proposed graph foundation model AnyGraph cannot work with heterogeneous graphs (graphs with multiple node types and edge types).
* The paper is very hard to read.
    * Figures do not have clear captions. Here are some examples:
        1. Figure 2 is not understandable.
        2. Where is Figure 4(c)? What does -Feat mean in Figure 4?
    * Notations are not well presented and explained. Here are some examples:
        1. In Section 3.1, how to get the entity representation like v_{c1}, v_{p1} is unclear. (I guess it is from equation 5).
        2. How do you define \hat{y}_{max} in equation 9?
    * Some tables do not have clear caption and explanation. Here are some examples
        1. Which model is the baseline in Table 2? Why only show the performance of Pubmed and Cora?
* Some of the design choices are not well justified and presented:
    * Why Adjacency matrices are used to init the node embeddings? The Adjacency matrices are used twice, one in node feature initialization and one in message passing.
    * What kind of reprocessing is applied to initial graph embeddings in feature and structure augmentation?
* The details of the evaluation result of each dataset corresponding to Table 1 is missing in the Appendix.
* I also want to see the comparison between AnyGraph and the baseline models trained with full training set (not the few-shot setting). This can help us to understand the gap between AnyGraph and full-training set training.
* How to choose the number of experts is unclear? Why using seven in the evaluation? What is the impact to use less experts and more experts?
* All the graph datasets used in evaluation are small graphs with maximum of 1M nodes. Can we also evaluate the performance of AnyGraph on large graphs like ogbn-products, ogbn-paper100M?
* The way how AnyGraph compute graph embeddings is similar to NetInfoF (https://arxiv.org/pdf/2402.07999). Can you compare the difference?

**Questions:**

* What is the inference overhead of AnyGraph?
* Why the approach consumes only 1/K memory resources during training or inference? You still need to host all the expert in memory.
* In Figure 3. With the increasing size of parameters, when AnyGraph’s performance improves first and then suddenly experiences a drop first and improves significantly. Why? Is there any explanation?
* How do you compute the training time of AnyGraph in Table3? Which training set are you using? Link1 or Link2?

---

### Official Review · Reviewer_B3K4 · 2024-11-04

**Soundness:** 3
**Presentation:** 3
**Contribution:** 2
**Rating:** 5
**Confidence:** 4

**Summary:**

The paper proposes a foundation graph model to handle graphs with (numerical) node attributes.

It used 38 diverse graph datasets for training.

It proposes a 'mixture of experts' model, thus reducing the number of parameters.

Experiments on these 38 datasets show better accuracy on link prediction and node classification, against popular baselines.

**Strengths:**

S1 nice idea, to use a mixture of experts

S2 also nice idea, to use of SVD for feature unification (Eq. 5), line 224, 225

S3 experiments show improvement on baselines

S4 nice ablation study

**Weaknesses:**

W1. too few datasets - hard to believe that the method may work on, say,
    patient-doctor-diagnoses graphs, that it has never seen before.

W2. Eq. 6, line243: adding the embeddings may lead to failures if the graph
    exhibits heterophily. Concatenation would be better - see, eg.,
    Table 1 of the SlimG paper
    [Yoo et al, KDD 2023] or https://arxiv.org/pdf/2210.04081

Minor, presentation suggestions:

M1. it would be nice to give the problem definition:
  given <??> find <??> to optimize <??>

M2. also, a table of symbols and definitions, like 'Flip()' on Eq5,
  $B$, $L'$, $K$,  on lines 291-294.

M3. what is the total wall-clock time for training, and for inference;
  and on what hardware (# CPUs, #GPUs, etc)
  Table 3 only gives training time per 100 steps.

**Questions:**

Rephrasing the 'weak' points above:

Q1 (based on W1): how can a practitioner be sure that AnyGraph will work for a graph outside the 3-4 groups (academic, social, etc) that were used in the paper

Q2 (based on W2): Does AnyGraph work well on heterophily graphs (see the datasets in Table 4, bottom, of the 'slimG' paper mentioned earlier)

Q3 (based on M3): what is the total wall-clock time (a) for training and (b) for inferencing - please

---

### Note · Authors · 2024-11-25

I have read and agree with the venue's withdrawal policy on behalf of myself and my co-authors.